# Feasibility and Acceptability of a Newborn Screening Program Using Targeted Next-Generation Sequencing in One Maternity Hospital in Southern Belgium

**DOI:** 10.3390/children11080926

**Published:** 2024-07-30

**Authors:** Tamara Dangouloff, Kristine Hovhannesyan, Davood Mashhadizadeh, Frederic Minner, Myriam Mni, Laura Helou, Flavia Piazzon, Leonor Palmeira, François Boemer, Laurent Servais

**Affiliations:** 1Neuromuscular Reference Center, Department of Pediatrics, University Hospital Liege, University of Liege, 4000 Liege, Belgium; tamara.dangouloff@uliege.be (T.D.); khovhannesyan@uliege.be (K.H.); davood.mashhadizadeh@guest.uliege.be (D.M.); flapiazzon@gmail.com (F.P.); 2Biochemical Genetics Lab, Department of Humans Genetics, CHU of Liege, University of Liege, 4000 Liege, Belgium; f.minner@chuliege.be (F.M.); myriam.mni@uliege.be (M.M.); laura.helou@chuliege.be (L.H.); lpalmeira@chuliege.be (L.P.); f.boemer@chuliege.be (F.B.); 3MDUK Oxford Neuromuscular Centre, Department of Paediatrics, University of Oxford, Oxford OX1 3QR, UK

**Keywords:** newborn screening, genomic, treatable disease, targeted next-generation sequencing

## Abstract

Purpose: Genomic newborn screening programs are emerging worldwide. With the support of the local pediatric team of Liege, Belgium, we developed a panel of 405 genes that are associated with 165 early-onset, treatable diseases with the goal of creating a newborn screening test using targeted next-generation sequencing for all early-onset, treatable, and serious conditions. Methods: A process was developed that informed the future parents about the project and collected their consent during a face-to-face discussion with a trained investigator. The first baby was screened on 1 September 2022. The main objective of the study was to test the feasibility and the acceptability of targeted sequencing at birth as a first-tier newborn screening approach to detect treatable genetic conditions or genetic conditions for which a pre-symptomatic or early symptomatic clinical trial is available. Results: As of 20 June 2024, the parents of 4425 children had been offered the test; 4005 accepted (90.5%) and 420 refused (9.5%). The main reasons for refusal were the research nature of the project and the misunderstanding of what constitutes genetic conditions. Conclusions: These data demonstrate the high acceptability of genomic newborn screening in a properly informed population.

## 1. Introduction

The diseases selected for newborn screening (NBS), which began in 1963, must meet the strict criteria established by Wilson and Jungner [1]: a severe, early-onset disease for which a treatment exists, whose natural history is well-known, and for which a reliable test exists. The current NBS programs test for between two and sixty-four severe treatable conditions [2] mainly using biochemical techniques. Recently, genetic tests for severe combined immunodeficiency disease and spinal muscular atrophy have been added to NBS programs in several countries [3,4,5,6]. Several severe and treatable conditions that match all the criteria are currently not included as no biomarkers are available. The rapid development of innovative therapies requires a mechanism to implement additional testing. Genomic newborn screening is increasingly recognized as a possible solution for these two unmet needs [7,8].

Several genomic NBS programs are currently underway [9]. The BabySeq Project in Boston screened nearly 300 children using exome sequencing between 2015 and 2017 [10]. In China, a study on biochemical screening and targeted gene panel sequencing for 128 conditions recruited 29,601 newborns in 2021 [11]. The Greek FirstSteps program, part of the BeginNGS program [12,13], aims to screen for 500 genetic diseases by genome sequencing. The Guardian Project, in New York, USA, screens for 450 genetic conditions that are not currently part of the standard newborn screening (https://pubmed.ncbi.nlm.nih.gov/35081954/, accessed on 26 January 2022).

Little is known about the acceptability of genomic newborn screening in the general population. The concerns in the general press [14,15] appear to be focused on cloning or the creation of databases with information that could be shared with insurance organizations [16] rather than the specific issue of NBS. The rare studies on the general extension of screening [17] or on specifically targeted diseases [18,19,20] indicate strong support from parents for genetic screening for treatable and even untreatable diseases [17]. Genomic NBS pilot programs are needed to evaluate the real-life acceptability of parents who have delivered a seemingly healthy baby. Pilot programs can provide more reliable information than a questionnaire. Recent data from a pilot program in Duchenne disease in the USA reported 85% agreement with genetic screening [21], and 40% to 80% agreement was reported in a spinal muscular atrophy pilot program in Germany [22].

To evaluate the acceptance of broad genomic screening, we developed a targeted next-generation sequencing (tNGS) program in Belgium, Baby Detect, which is a prospective observational pilot study of population newborn screening designed to evaluate the acceptability, reliability, and feasibility of first-tier tNGS in southern Belgium to facilitate early diagnosis and treatment.

## 2. Materials and Methods

### 2.1. Ethical Approval

This project was discussed with the local ethics committee and approved (n° 2021/239) in accordance with the Declaration of Helsinki. The project was officially launched on 1 September 2022 in the maternity and neonatal units of the Citadelle Hospital in Liege, Belgium. The study is registered as NCT05687474 (clinicaltrial.gov).

### 2.2. Gene Selection

The disease selection criteria in Baby Detect were based on the principles of Jungner and Wilson [1,23], with the precision of the following fundamental principles:Significant life-expectancy consequences or severe disability associated with untreated patients.Disease onset in childhood (i.e., before 5 years).Strong genotype–phenotype correlation.Existence of a disease-modifying treatment or access to a clinical trial for pre-symptomatic or early symptomatic stage.Significant benefit of early treatment.Endorsement by treating pediatricians from CHU of Liege.

A list of genes was prepared by the investigators, and genes were discussed individually with physicians in appropriate subspecialties of the pediatric department of CHU Liege. The final list included 360 genes (Appendix A). Using the same methodology of refinement, a second panel of 405 genes (14 were deleted and 63 were added) was designed after 1 year of screening (Appendix A).

### 2.3. Enrollment

The study is advertised through press releases to mainstream media and through social network announcements and by targeted information to pregnant women. There is no cost to parents who participate. Information to the parents is provided during pregnancy and shortly after birth through leaflets, videos, a website (https://babydetect.com/en/, accessed on 30 July 2024), and verbal explanations by gynecologists, pediatricians, and midwives (Figure 1). Parents are provided links to short videos viewable on smartphones. The day after delivery, in the maternity ward or the neonatal unit, almost all parents are invited by a trained investigator to participate in the study. Reasons for refusal are recorded when clearly expressed. The ethnicity of parents accepting or refusing participation cannot be collected due to regulatory and ethical reasons. Consent from both parents is sought, but, if only one parent is available, his or her consent alone is accepted. Electronic consent forms are available in the three official Belgian languages (French, Flemish, and German). Leaflets and the information consent form are additionally available in the eight other most spoken languages in southern Belgium (Arabic, English, Italian, Polish, Romanian, Russian, Spanish, and Turkish). The website with the explanation about the study, videos (one complete and a dozen very short ones related to a specific question), and the description of all the diseases included in the panel are available in French and in English.

Parents who consented are additionally asked to consent that the remaining DNA be anonymously stored and used for research purposes. After 1 year, an amendment was introduced to ask parents to agree to be recontacted when the child is one year old to request information on parents’ opinion of their research participation.

The hospital where the study was conducted is a public institution in southern Belgium that serves an ethnically diverse population. The maternity unit registers around 2500 births a year. Parents from outside the hospital can apply to take part in the study.

### 2.4. Technical Aspects

After obtaining parental consent, a sample of blood was collected on a dried blood spot (DBS) yellow-colored card in addition to the standard DBS sample between 48 and 96 h after birth. The laboratory workflow began with DNA extraction from the DBS card (QIAamp DNA Investigator Kit, Qiagen, Hilden, Germany) followed by tNGS (Twist Bioscience, South San Francisco, CA, USA) and Illumina sequencing (NovaSeq 6000, NextSeq 550, San Diego, CA, USA). Raw sequencing data were demultiplexed using an in-house bioinformatics pipeline that allows variant calling. Interpretation of variants and reporting were performed using Alissa Interpret software version 5.4.2 (Agilent, Santa Clara, CA, USA). The workflow approach was validated for manual operation and pilot screening taking into account acceptable standards for diagnostic settings. Raw data were analyzed using a custom-developed bioinformatics pipeline for SNP and INDEL inference. For variant interpretation and reporting (class 5 and 4 SNVs), a decision tree (Alissa Interpret, Agilent) was validated on 600 positive and negative samples.

### 2.5. Results Reporting

As with conventional neonatal screening in Belgium, if the results are negative, no action is taken. Parents are not informed of a negative result. In case of an abnormal result, the parents are contacted by the disease specialist at the reference center to arrange a face-to-face consultation at the hospital as soon as the specialist deems it necessary. At this meeting, the results are discussed, and an independent blood sample is taken for further diagnostic confirmation.

## 3. Results

From 1 September 2022 to 20 June 2024, of the 4472 births at our hospital, the parents of 4425 infants (99%) were provided information on Baby Detect (Figure 1). Of these, the parents of 4005 (90.5%) provided consent, and the parents of 420 (9.5%) refused, for an 89.6% participation rate. The most common reasons for refusal are listed in Table 1.

The parents of 93% of the infants who consented to the screening also consented to DNA storage and anonymous use for research purposes, and 86% agreed to be recontacted after 1 year. Most (91%) of the consents were gathered within 3 days of birth; the remaining were obtained within 1 month (median 1.41 days; IQ 1.33 days). Of the infants screened, 2031 (51%) were male and 1974 (49%) were female. The median weight of the infants was 3250 g (525 g–5720 g), and the gestational age was 39 weeks (24–42 weeks).

**Figure 1 children-11-00926-f001:**
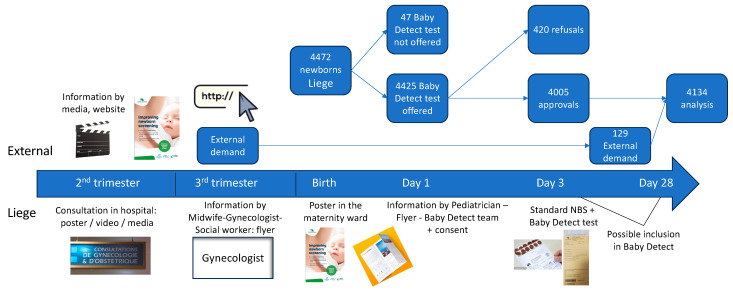
Flowchart of information presentation, consent, and testing during Baby Detect pilot.

## 4. Discussion

Our study demonstrates the high level of acceptability of tNGS-based NBS within the Belgian population. The 90.5% acceptance rate, obtained in the context of an opt-in research project, compares favorably with pilot programs worldwide. For instance, the acceptance of spinal muscular atrophy pilot screening in Germany increased from 40% to 87% in the second year. Conversely, the BabySeq program reported low acceptance of only 6.9%, which may have been due to the requirement of attendance at an informational meeting and a subsequent return to the hospital for the results [10]. This disparity emphasizes the importance of effective communication and program logistics, such as minimizing the need for parental return visits and maximizing document comprehension. For example, we provide information in languages spoken by traditionally underrepresented ethnicities in Liege, where a substantial portion of the population comprises non-Belgians (18% in 2013) [24]. Additionally, the dedicated staff trained in the consent procedures may have played a pivotal role in achieving high participation rates. Specifically trained students were responsible for explaining the study and obtaining consent from the parents. Importantly, prior communication from trusted physicians (pediatricians and gynecologists) significantly eased the explanation process. Finally, downloadable videos that addressed the frequently asked questions were appreciated by young mothers.

The psychological impact on parents of an early announcement of positive screening results is important to assess, and we plan to monitor all the newborns identified with a disease through Baby Detect, both from a medical point of view and in terms of the parents’ quality of life.

Genomic newborn screening presents a range of significant opportunities, including the ability to expand the scope of detectable conditions, quickly add new conditions at low incremental costs, and improve access to precision treatments for rare diseases. These advantages apply particularly to genome and exome sequences more than to tNGS. Establishing the gene list for genomic NBS presents a multifaceted challenge, particularly in navigating the complexities of genomic data analysis and interpretation within the context of newborn health. Unlike post-symptomatic genetic testing, where variant interpretation often relies on detailed phenotypic data and parental genomic information, newborns generally do not express any phenotype at the time of sampling, making phenotype-driven interpretation useless. Genomic NBS interpretation is therefore centered on variant properties, population data, and the existing reports of variant pathogenicity. Additionally, the issue of variable penetrance and expressivity complicates the interpretation, particularly for conditions with varying clinical presentations and genotype–phenotype correlations. For this reason, in Baby Detect, we only communicate the variants with high penetrance. Moreover, designing genomic NBS programs necessitates careful consideration of the balance between identifying severe treatable disorders and minimizing false-positive results—as the risk of false positives is cumulative with the number of genes that are tested—as well as addressing the broader healthcare system implications and access to treatment.

As of this submission, over 4005 children have undergone screening through Baby Detect, and the program is ongoing. We will report separately on the cases identified. Numerous challenges remain, particularly regarding the gene list, which must continually evolve in close collaboration with pediatricians. We have chosen to empower the clinicians overseeing the identified children with the final say in selecting the genes to be tested. It is essential to offer these physicians the option to suggest new genes for inclusion and to accommodate any preferences for gene removal. The next hurdle will be expanding the screening program to other hospitals, which presents technical difficulties, such as the need for automation in the sample analysis processes. Moreover, securing sufficient financial resources will be crucial for widespread expansion, necessitating considerations such as sponsorship, government support, or involvement from start-up ventures.

Our study has yielded valuable insights into the feasibility and acceptability of genomic newborn screening. One of the limitations is that the studied population may not fully represent the broader demographic diversity of Belgium or Western Europe, potentially introducing biases in the results. Indeed, the maternity units belong to a public hospital with a very diverse and potentially underserved population. The fact that the test was offered free of charge but in the context of a study may also bias the results in two ways. On one hand, a fully free-of-charge brand-new test can be attractive. On the other hand, the research nature of the project was one of the identified causes of refusal (“I don’t want any research to be conducted on my baby”). These data should be confirmed in a broader population and outside the context of a research program. Longitudinal studies tracking the long-term health outcomes of screened infants and evaluating the efficacy of early interventions will provide valuable data to inform future program enhancements and policy decisions. An ancillary study has already been established to follow patients who have been identified through the program.

In conclusion, we demonstrate the high acceptability of genomic-based NBS in southern Belgium. This high acceptability appears to depend on the perception of information sharing as an ongoing process that utilizes appropriate tools and language for the intended audience.

## Figures and Tables

**Table 1 children-11-00926-t001:** Reasons for refusal.

Wording/Reasons for Refusal	Number of Refusals
No reason	225
Family in good health/eldest child in good health/pregnancy test normal/child has been examined by pediatrician and is in good health: not necessary	37
Only what is mandatory/no extras/20 illnesses = enough	32
Father does not want	19
Did not sign at maternity hospital (forgot) Oral agreement, but would not sign after phone call	17
The fact that it is a study/experimental/that there is no hindsight/does not want her child to be a mouse in a laboratory/if consent required = risk	16
Child too small, too many blood tests, painful	11
Stress (delay/many illnesses)	11
Fear	10
Language	10
Not conducted for older child	8
Fatality/accepting one’s fate	8
Fear of false positive	3
Prefers to wait until baby is ill	3
Anti-vaccine/Anti-COVID-19: conspiracy if illness not visible	3
Illnesses already present in child or siblings	2
Fear of data storage	1
Fear of genetic testing	1
Too depressed to risk receiving bad news	1
Does not want to be categorized as ill	1
The human body has to fight	1

## Data Availability

The original contributions presented in the study are included in the article and Appendix A, further inquiries can be directed to the corresponding authors: tamara.dangouloff@uliege.be.

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
