# Peer review of "Feasibility and Acceptability of a Newborn Screening Program Using Targeted Next-Generation Sequencing in One Maternity Hospital in Southern Belgium"

_children, 2024, doi:10.3390/children11080926_

Round 1

Reviewer 1 Report

Comments and Suggestions for Authors

This short report deals with an expanded new born screening program using targeting sequencing in Liege, Belgium.  The data reported solely deal with the acceptance of participation by the parents of the newborns; actual results of mutations found in the screened genes are not described. Therefore, this constitutes a very limited report. The title is misleading, this is not entire Southern Belgium, or as I first thought Wallonia, it is a single hospital, hence a single centre study. This should be clearly indicated.

Next, authors find the 90% acceptance rate high, but there is no comparison with other studies, so why is this a high percentage?  Compared to most childhood vaccination is toddlers, this is a relatively low percentage.  Hence a comparison with other mild invasive procedures in babies / toddlers should be included

Reviewer 2 Report

Comments and Suggestions for Authors

Reviewer’s Report

Title: Feasibility and Acceptability of a Newborn Screening Program Using Targeted Next-Generation Sequencing in Southern Belgium

The manuscript addresses an important and topic in the field of genomic medicine. The study provides valuable data on understanding of the genomic screening programs.

Specific Comments:

* Please include numerical data on the acceptance rate and the number of children in the abstract.

* Could you provide more details on the validation of the tNGS workflow, as well as the criteria for variant interpretation?

* What are potential limitations of the study? I would mention the lack of long-term follow-up data. What about potential demographic biases?

Round 2

Reviewer 1 Report

Comments and Suggestions for Authors

No further comments

Author Response

Dear reviewer, 

We have taken into account the comments made by the editor and the two reviewers in order to improve this article as much as possible. The methodology will be described more specifically in a dedicated article, focusing on a precise description of both the laboratory workflow (extraction / tool used) and the bioinformatics and interpretation of variants. Here, we focus on describing the process and acceptability of this screening. 
